# The Chronic Toxicity of Endocrine-Disrupting Chemical to *Daphnia magna*: A Transcriptome and Network Analysis of TNT Exposure

**DOI:** 10.3390/ijms25189895

**Published:** 2024-09-13

**Authors:** Jun Lee, Hyun Woo Kim, Dong Yeop Shin, Jun Pyo Han, Yujin Jang, Ju Yeon Park, Seok-Gyu Yun, Eun-Min Cho, Young Rok Seo

**Affiliations:** 1Institute of Environmental Medicine for Green Chemistry, Department of Life Science, Biomedi Campus, Dongguk University, 32 Dongguk-ro, Ilsandong-gu, Goyang-si 10326, Republic of Korea; 2Department of Nano, Chemical & Biological Engineering, College of Natural Science and Engineering, Seokyeong University, Seoul 02173, Republic of Korea

**Keywords:** endocrine-disrupting chemical, chronic exposure, gene expression profile, biological network analysis, adverse outcome pathway (AOP)

## Abstract

Endocrine-disrupting chemicals (EDCs) impair growth and development. While EDCs can occur naturally in aquatic ecosystems, they are continuously introduced through anthropogenic activities such as industrial effluents, pharmaceutical production, wastewater, and mining. To elucidate the chronic toxicological effects of endocrine-disrupting chemicals (EDCs) on aquatic organisms, we collected experimental data from a standardized chronic exposure test using *Daphnia magna* (*D. magna*), individuals of which were exposed to a potential EDC, trinitrotoluene (TNT). The chronic toxicity effects of this compound were explored through differential gene expression, gene ontology, network construction, and putative adverse outcome pathway (AOP) proposition. Our findings suggest that TNT has detrimental effects on the upstream signaling of Tcf/Lef, potentially adversely impacting oocyte maturation and early development. This study employs diverse bioinformatics approaches to elucidate the gene-level toxicological effects of chronic TNT exposure on aquatic ecosystems. The results provide valuable insights into the molecular mechanisms of the adverse impacts of TNT through network construction and putative AOP proposition.

## 1. Introduction

Endocrine-disrupting chemicals (EDCs) are toxic compounds that impede the functions of the endocrine system, affecting not only the synthesis, secretion, and metabolism of hormones but also biological processes (BPs) related to growth, development, and cancer [1,2,3]. The use of EDCs has witnessed a notable surge in the industrial, agricultural, and pharmaceutical fields in recent decades, posing a substantial threat to animal life [4,5,6]. In aquatic environments, EDCs can occur naturally, but their concentrations can also increase due to human activities, including industrial activities, agricultural runoff, mining, and the discharge of domestic and municipal wastewater [7,8,9,10]. Diverse adverse effects of EDCs have been observed in numerous aquatic field and laboratory studies involving freshwater fauna [7].

2,4,6-trinitrotoluene (TNT) has been detected in water sources proximal to military installations [11]. This contamination is attributed to the effluent discharge from explosive-manufacturing facilities, which facilitates the introduction of TNT into aquatic ecosystems. TNT contamination was reported in Sweden, where a concentration of 3 mg/L of TNT was observed in ponds near to industrial facilities [12]. Numerous studies have been conducted on the removal of TNT from aquatic environments as well as on the toxicity resulting from TNT exposure [13,14,15,16]. TNT, a potent explosive, can form π complexes with estrone, a female sex hormone characterized by an electron-rich aromatic ring. Frontier molecular orbital analyses, including HOMO and LUMO, demonstrate the π complex’s remarkable resistance to oxidation, suggesting a potential interference of TNT with estrone’s biological functions [17]. Furthermore, animals such as *Mus musculus*, *Rattus rattus*, and *Caenorhabditis elegans* treated with TNT experience adverse effects on their reproductive systems [18,19,20,21]. This evidence points to the broader toxicological implications of TNT, particularly its impact on sexual health through the disruption of hormonal activities, underscoring the necessity for comprehensive studies on its biological effects [17].

The water flea, a kind of crustacean zooplankton, is ubiquitous in both lotic and lentic ecosystems [22]. Hence, the impact of toxicants on reproduction and population dynamics in the water flea could exacerbate disruptions to the overall health of freshwater ecosystems [23]. *Daphnia magna* (*D. magna*), a species of water flea, serves as a recommended model organism in the U.S. EPA guidance (OPPTS 850.1300) and OECD test guidelines (No. 211), presenting significant advantages for genetic research [24,25]. In controlled environmental conditions, it exhibits a parthenogenetic reproductive cycle, facilitating the generation of genetically homogeneous individuals [26]. Furthermore, *D. magna* is a suitable individual for reproductive research because it has a short life cycle, good reproductive ability, and a size that is easy to observe with the naked eye [22]. Considering its position in the ecological hierarchy, *D. magna* constitutes an essential genus that can be valuable in evaluating ecotoxicity, owing to its unique life cycle that offers numerous sensitive endpoints for measuring physiological activities [23,27].

Public databases, including the Gene Expression Omnibus (GEO), offer a wealth of detailed biological data, including gene expression profiles of numerous mRNAs at a particular moment in time, and changes in the mRNA levels can be used to detect the cellular response to specific conditions, including medications or toxic substances [28,29,30].

Diverse bioinformatics analyses, such as differentially expressed gene (DEG) analysis, functional enrichment analysis, biological network analysis, and suggesting putative adverse outcome pathways (AOPs), have been used to understand the biological functions of gene expression profiles. Gene ontology (GO) categorizes genes based on their biological process (BP), cellular component (CC), and molecular function (MF) to elucidate biological events at the cellular and molecular levels. A biological network offers an effective approach for comprehensively identifying the adverse effects induced by TNT on cellular processes, potential phenotypes, and central genes. AOP is one of the biological mechanism explanation methods that elucidate a series of biological events stemming from exposure to substances, including EDCs, until an adverse outcome is reached.

In this study, we explore the toxic effects of TNT chronic exposure through diverse bioinformatic analyses such as differentially expressed gene (DEG) analysis, functional enrichment analysis, and biological network analysis, and suggest putative adverse outcome pathways (AOPs) to understand the biological events at the cellular level, the adverse effects, and biological mechanisms. Finally, we present a comprehensive approach to understand these mechanisms in aquatic organisms, which helps grasp the broader environmental health impacts of EDCs.

## 2. Results

### 2.1. Fucntional Enrichment Analysis

From transcriptome data following chronic exposure to TNT, 1759 DEGs were identified, with 688 upregulated and 1071 downregulated. Functional enrichment analysis based on their expression patterns was performed, and the top 10 upregulated and downregulated BPs, CCs, and MFs were identified (Table 1). Metabolism-related GO terms predominated in both upregulated and downregulated BPs, with ‘Cellular process’ exhibiting the lowest FDR in both categories. Upregulated CCs showed a prevalence of GO terms related to cellular organelles and membrane-bounded organelles. Conversely, downregulated CCs were notably associated with myofibers. Upregulated MFs involved the binding of substances other than proteins, while downregulated MFs included oxidative stress-related GO terms like ‘Oxidoreductase activity’ and ‘Glutathione transferase activity’.

### 2.2. Comprehensively Functional Enrichment Analysis for DEGs

Irrespective of their expression patterns, DEGs were integrated for GO analysis, confirming associations with the endocrine system, development, and reproduction (Table 2). In the context of endocrine function, associations emerged between five BPs related to development and eight BPs related to reproductive functions. Two key ontological categories were highlighted: ‘Organic substance biosynthetic process’, linked to hormone synthesis, and ‘Organic substance transport’, associated with hormone transport.

### 2.3. Toxicity Mechanisms of Chronic TNT Exposure in D. magna

#### 2.3.1. Biological Network Analyses

To elucidate the reproductive toxicity mechanisms of chronic TNT exposure, DEGs were screened through interactions of transcribed protein from each DEG (Figure 1a). A biological network shows that chronic TNT exposure could cause adverse effects on reproduction and development in *D. magna* (Figure 1b). The biological network demonstrates that abnormal phenotypes in development, including body size, fertility, and oxidative stress response, can result from dysfunction in various cellular processes, such as reproductive processes, female gamete generation, and germ cell development. In the biological network, LOC116919267 (SUMO-conjugating enzyme UBC9), LOC116919853 (eukaryotic initiation factor 4A-III), LOC116919548 (transformer-2 protein homolog α), and LOC116923253 (histone deacetylase 1) exhibited high degree and betweenness centrality and were selected as central genes. An endocrine-focused biological network was found based on these five key genes and the interaction between central genes, central genes–cellular processes, and central genes–phenotypes (Figure 1c). The central genes are related to specific cellular processes involved in reproductive and developmental functions. In the endocrine-focused biological network, reproductive- and development-related adverse effects were mainly observed, and female reproduction-related adverse effects were also detected.

#### 2.3.2. Putative Adverse Outcome Pathway (AOP) Development for TNT

A putative adverse outcome pathway (AOP) was proposed, hypothesizing that a putative molecular initiating event (MIE) occurs due to chronic TNT exposure, followed by changes in the expression of casein kinase 1, epsilon (CK1ε), and casein kinase II (CK2) (Figure 1d). The differential expression of CK1ε and CK2 leads to the formation of the segment polarity protein dishevelled (Dvl) complex. This complex interacts with and regulates the activity of the transcription factors T-cell factor/lymphoid enhancer (Tcf/Lef). The formation of the Dv1 complex promotes Tcf/Lef to bind to DNA and the transcription of genes crucial for oocyte maturation and early development. The deregulation of these transcription factors results in the activation of genes that should otherwise be tightly controlled, leading to potential developmental and reproductive abnormalities in *D. magna*.

## 3. Discussion

The DEGs derived from the transcriptome of the whole body of *D. magna*, individuals of which were exposed to TNT for 21 days, were found. The biological function of DEGs were analyzed using functional enrichment analysis, biological network analysis, and putative AOP [31]. At the cellular level, chronic exposure to TNT results in unusual metabolism, abnormal cell anatomical structures, reduced offspring development, and altered binding processes (Table 1). Among the upregulated BPs, ‘Cellular process’, ‘Biological regulation’, and ‘Cellular component organization or biogenesis’ were featured. In contrast, downregulated BPs featured ‘Organonitrogen compound metabolic process’, ‘Response to stimulus’, ‘Regulation of biological quality’, and ‘Response to ethanol’. The chemical structure of TNT dictates that biological detoxification pathways predominantly follow the reductive transformation of the nitro groups, making TNT recalcitrant to environmental mineralization. In the context of CCs, GO terms like ‘Intracellular anatomical structure’, ‘Cellular anatomical entity’, and ‘Cytoplasm’ were recurrent. For MFs, ‘Binding’, ‘Protein binding’, and ‘Catalytic activity’ were consistent GO terms across both upregulated and downregulated categories. Upregulated MFs involved the binding of substances other than proteins, while downregulated MFs included oxidative stress-related GO terms like ‘Oxidoreductase activity’ and ‘Glutathione transferase activity’. Both oxidoreductase and reduced glutathione are major antioxidants and play a key role in maintaining redox homeostasis [32,33,34]. The GO term ‘Organic substance biosynthetic process’ was linked to hormone synthesis, and ‘Organic substance transport’ was associated with hormone transport.

To comprehend the cellular functions of DEGs, functional enrichment analysis was conducted regardless of their expression patterns. After validating the GO terms with significant relevance to reproduction (FDR < 0.05), a total of 14 GO terms were recognized within BPs, and 1 GO term was identified each in CCs and MFs (Table 2). Notably, the BPs were related to development, reproduction, and the metabolism of organic substances, while GO terms pertinent to cell division were discovered in CCs, and GO terms related to organic compound binding were found in MFs. Additionally, five BPs related to development were elucidated, along with eight BPs associated with reproductive processes. Notably, the GO terms ‘Developmental process involved in reproduction’ and ‘Germ cell development’ were both linked to reproduction. Two reproductive-related BPs involved the genesis of germ cells, and three were related to meiotic cell division. Observations of ‘Female gamete generation’ and ‘Female meiotic nuclear division’ in *D. magna* highlight the adverse effects of TNT on reproductive functions. Additionally, ‘Meiotic spindle’ and ‘Organic cyclic compound binding’ were identified in CCs and MFs, respectively, linking them to reproduction and hormone binding.

Through biological network analysis, we achieved a comprehensive understanding of the chronic toxicity of TNT (Figure 1b). It was possible to identify adverse effects on putative phenotypes associated with reproduction, development, and body size. In the endocrine-focused biological network (Figure 1c), adverse effects on reproduction-related phenotypes and muscle development were observed, indicating that genes LOC116919267 (SUMO-conjugating enzyme UBC9), LOC116919853 (eukaryotic initiation factor 4A-III), LOC116919548 (transformer-2 protein homolog α), and LOC116923253 (histone deacetylase 1) could play crucial roles in the chronic toxicity mechanism of TNT. The main functions of UBC9 include influencing protein localization within the cell, altering protein–DNA interactions, and modifying protein–protein interactions [35,36]. It is involved in key cellular processes, such as cell cycle regulation, stress response, and development, notably in oocytes and embryos, indicating its role in transcriptional activation and possibly chromatin remodeling [37]. Furthermore, it is concerned with muscle development: the knockdown of endogenous nhp2l1 in zebrafish disrupts skeletal muscle development [38]. Eukaryotic translation initiation factor 4A III (*EIF4A3*) regulates post-transcriptional gene expression by aiding in precursor mRNA splicing and influencing nonsense-mediated mRNA decay. It also supports the expression of important selenoproteins, such as phospholipid hydroperoxide glutathione peroxidase and thioredoxin reductase 1 [39]. Transformer 2 α homolog (*TRA2A*) promotes growth, movement, and chemotherapy resistance by regulating specific splicing events independent of other splicing factors [40]. Additionally, cell proliferation is tightly regulated by cyclins, with RNA processing factors being the most abundant functional class of cyclin-associated proteins, including TRA2A in human cells [41,42]. TRA2A is overexpressed in glioma tissues, enhancing the proliferation, migration, invasion, and EMT of glioma cells [43]. Histone deacetylase 1 is involved in RNA translation and protein synthesis [44].

For intuitive insights into the chronic toxic mechanism of TNT, we proposed a putative AOP based on the KEGG pathway database [45]. The putative AOP showed that TNT induces changes in the expression of Ck1ε and Ck2, potentially leading to the occurrence of abnormal transcription of target genes, such as those involved in oocyte maturation and early development (Figure 1d). This suggests that chronic exposure to TNT affects reproduction and reproductive hormone metabolism. The differential expression of Ck1ε and Ck2 leads to the formation of the Dvl complex. This complex interacts with and regulates the activity of Tcf/Lef transcription factors [46]. Dvl is activated when Wnt binds to its receptor Frizzled (Fz) and coreceptor Lrp5/6 on the cell surface [47]. This binding event is crucial for initiating the Wnt signaling cascade. When Wnt binds to its receptors, Dvl is phosphorylated by Ck1ε, which inhibits the Gsk-3β-mediated phosphorylation of β-catenin [48,49]. This prevents β-catenin degradation, allowing it to accumulate in the cytoplasm and translocate to the nucleus to activate gene transcription [50]. However, the formation of the Dvl complex due to altered kinase activity disrupts this inhibition, allowing Tcf/Lef to bind to DNA and promote the transcription of genes crucial for oocyte maturation and early development. The deregulation of these transcription factors results in the activation of genes that should otherwise be tightly controlled, leading to potential developmental and reproductive abnormalities in *D. magna*.

Understanding these mechanisms in *D. magna* provides valuable insights into potential risks and pathways of chronic exposure to TNT, which could adversely affect aquatic organisms’ health, especially for populations reliant on aquatic ecosystems for food and water resources. Research on specific genes and mechanisms through diverse bioinformatic analysis methods could provide a comprehensive understanding of these complex mechanisms and new insights into exploring the toxicity of chemicals in aquatic organisms. Since our analysis was conducted using only partial data, more public genomic data and experiments need to be integrated. The interaction between *D. magna* genes was explained using the interactions of orthologous genes in other species, and gene–phenotype associations were also substituted with phenotypes of orthologous genes in *Drosophila melanogaster* (*D. melanogaster*) due to a lack of research on gene–gene and gene–phenotype interaction on *D. magna*. However, our study utilized only a single dataset of chronic TNT exposure. To fully understand the chronic toxic effects of TNT, a broader range of datasets is necessary. Additionally, future research should focus on examining the recovery of gene expression patterns modulated by chronic exposure to provide more accurate information for TNT regulation. Such studies would offer valuable insights into the persistence of genetic alterations induced by chronic TNT exposure and the potential for recovery at the molecular level. This information would be crucial for developing more effective regulatory strategies and understanding the chronic ecological impacts of TNT contamination.

## 4. Materials and Methods

### 4.1. Data Collection and Differentially Expressed Gene Screening

TNT chronic exposure gene expression profile datasets were obtained from the GEO database (https://www.ncbi.nlm.nih.gov/geo/, accessed on 3 October 2023). The gene expression dataset (GSE43960) includes the whole body of *D. magna* chronically exposed to TNT. The dataset was generated through microarray analysis following exposure to TNT of *D. magna* at a concentration of 1.85 mg/L for 21 days, in accordance with the guidance outlined by the U.S. EPA (OPPTS 850.1300), as implemented by Stanley et al. [51]. The exposure concentration was selected based on the dose–response relationship observed in preliminary range-finding experiments, according to the U.S. EPA guidelines (OPPTS 850.4200) [52]. TNT exposure was renewed three times a week, with one *D. magna* added per test chamber across 10 test chambers per treatment level, while the control exposure used solvent. The DEGs were screened with |Fold Change| ≥ 1.5 and *p*-value < 0.05 using the limma package (ver. 3.58.1) in R software [53].

### 4.2. Functional Enrichment Analysis

Gene ontology enrichment analysis was conducted using STRING database (https://string-db.org/, accessed on 2 January 2024) and false discovery rate (FDR) < 0.05 was considered to be statistically significant.

### 4.3. Network Analysis

#### 4.3.1. Protein–Protein Interaction Analysis

The protein–protein interaction data were sourced from the STRING database for *D. magna* DEGs, with an interaction score threshold of 0.4 [54,55].

#### 4.3.2. Gene–Phenotype Association Analysis

To find associations between genes and phenotypes, transcribed proteins of DEGs were translated into their *D. melanogaster* which are evolutionarily close to *D. magna* species using NCBI BLASTp [56,57]. The association between translated orthologous gene and phenotypes were found in FlayBase (https://flybase.org/, accessed on 30 June 2024).

#### 4.3.3. Biological Network Analysis and Endocrine-Focused Biological Network Analysis

Biological networks were constructed incorporating protein–protein interaction information, cellular processes, and orthologous phenotypes correlations. Top genes in the endocrine-focused biological network were identified by considering their degree and betweenness centrality. The cellular processes and phenotypes in the endocrine-focused biological network are connected to the selected genes, showing high degree and betweenness centrality.

### 4.4. Putative Adverse Outcome Pathway (AOP) Development

The putative AOP for chronic exposure to TNT in *D. magna* was analyzed based on the Wnt signaling pathway in the Kyoto Encyclopedia of Genes and Genomes (KEGG) database.

## 5. Conclusions

EDCs present significant risks to both the environment and living organisms by disrupting hormone functions, which can affect growth and development. The increased use of EDCs in industrial, agricultural, and pharmaceutical sectors has led to their accumulation in aquatic environments due to wastewater discharge. Various studies have shown that TNT can cause harmful effects in freshwater organisms, potentially disrupting ecosystems. *Daphnia magna*, a species of water flea, is an important model organism for studying the impact of toxic substances due to its key role in aquatic food webs and its ease of study. Data from gene expression profiling, such as those available from the GEO, provide valuable information on how chronic exposure to EDCs affects gene expression in this organism. We focused on datasets related to the exposure of *D. magna* to TNT, revealing significant alterations in gene expression linked to metabolism, cellular processes, and reproductive functions. Exposure to TNT was shown to interfere with the Wnt signaling pathway, leading to reproductive and developmental abnormalities in *D. magna*. Central genes such as LOC116919267 (SUMO-conjugating enzyme UBC9) and LOC116919853 (eukaryotic initiation factor 4A-III) were identified as central to the toxic effects observed.

By using techniques such as functional enrichment analysis and constructing biological networks, we were able to identify AOPs that describe the sequence of biological events linking EDC exposure to negative health effects. Understanding these mechanisms in aquatic organisms helps grasp the broader environmental health impacts of EDCs, especially for communities that depend on aquatic ecosystems for food and water. The importance of transcriptomic and network analyses in elucidating the complex interactions between environmental pollutants and biological systems was shown. Further research is essential to fully understand the implications of EDCs and to develop strategies for mitigating their impact on aquatic ecosystems and human health.

## Figures and Tables

**Figure 1 ijms-25-09895-f001:**
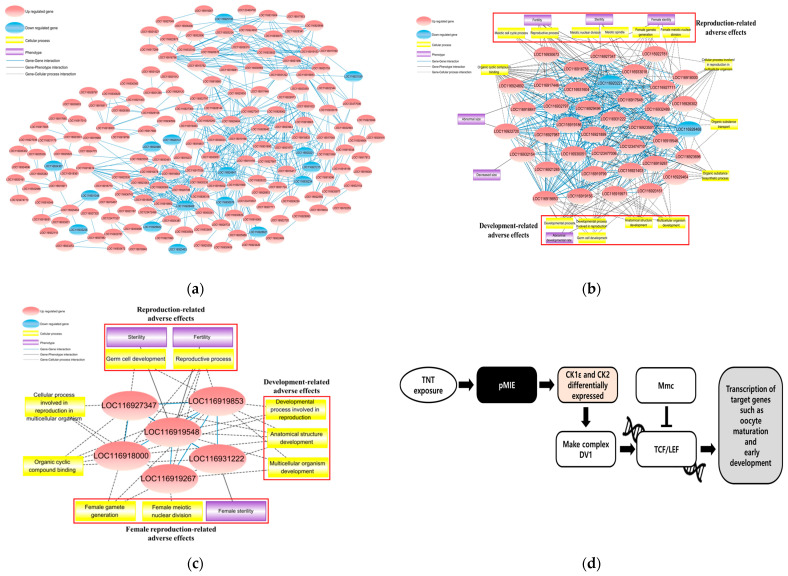
Biological networks and putative AOP for TNT chronic toxicity in *D. magna*. (**a**) Interactions between all DEGs; the interactions represent relationships between translated proteins of each DEG. (**b**) Biological network for DEGs associated with chronic exposure to TNT. This biological network shows that reproduction- and development-related adverse effect are strongly associated; (**c**) Endocrine-focused biological network for DEGs associated with chronic exposure to TNT. This endocrine-focused biological network demonstrates strong associations with reproduction, development, and adverse effects related to female reproduction. The red ovals represent upregulated genes of *D. magna*, while blue ovals indicate downregulated genes of *D. magna*. The yellow box and purple box indicate cellular process and phenotype respectively. The blue connecting lines indicate biological association between genes. The gray line and dashed gray line indicate gene–phenotype interaction and gene–cellular process interaction, respectively. (**d**) A putative AOP for chronic TNT exposure. This putative AOP demonstrates Ck1ε and Ck2 are differentially expressed after a putative molecular initiating event (pMIE), and the transcription of target genes which play roles in oocyte maturation and early development is regulated after Tcf/Lef regulation.

**Table 1 ijms-25-09895-t001:** The top 10 gene ontology terms for upregulated DEGs and the top 10 GO terms for downregulated DEGs from *D. magna* chronically exposed to TNT.

GO Term Category	GO Term for Upregulated DEGs		GO Term for Downregulated DEGs	
	GO Term Description	FDR	GO Term Description	FDR
Biological process	Cellular process	5.42 × 10^−90^	Cellular process	9.71 × 10^−17^
	Organic substance metabolic process	9.54 × 10^−51^	Metabolic process	4.73 × 10^−11^
	Metabolic process	9.54 × 10^−51^	Organic substance metabolic process	9.28 × 10^−10^
	Nitrogen compound metabolicprocess	2.1 × 10^−49^	Organonitrogen compoundmetabolic process	1.26 × 10^−8^
	Primary metabolic process	1.29 × 10^−48^	Response to stimulus	4.84 × 10^−8^
	Macromolecule metabolic process	1.37 × 10^−46^	Primary metabolic process	1.23 × 10^−6^
	Cellular metabolic process	1.15 × 10^−44^	Nitrogen compound metabolic process	1.40 × 10^−5^
	Biological regulation	3.36 × 10^−43^	Regulation of biological quality	3.83 × 10^−5^
	Cellular component organization or biogenesis	3.99 × 10^−40^	Glutathione metabolic process	6.61 × 10^−5^
	Regulation of biological process	6.02 × 10^−38^	Response to ethanol	1.10 × 10^−4^
Cellular Component	Intracellular anatomical structure	1.86 × 10^−104^	Cellular anatomical entity	3.54 × 10^−23^
	Cellular anatomical entity	1.15 × 10^−99^	Cytoplasm	6.45 × 10^−8^
	Intracellular organelle	1.63 × 10^−69^	Sarcomere	4.75 × 10^−6^
	Organelle	7.16 × 10^−69^	Intracellular anatomical structure	5.18 × 10^−6^
	Intracellular membrane-boundedorganelle	1.29 × 10^−58^	Extracellular region	6.61 × 10^−6^
	Membrane-bounded organelle	4.02 × 10^−56^	Supramolecular fiber	1.10 × 10^−4^
	Protein-containing complex	1.23 × 10^−53^	Membrane	1.10 × 10^−4^
	Cytoplasm	2.31 × 10^−52^	Z disc	2.30 × 10^−3^
	Nucleus	2.24 × 10^−50^	Organelle	2.90 × 10^−3^
	Ribonucleoprotein complex	6.20 × 10^−34^	Intracellular organelle	7.60 × 10^−3^
Molecular function	Binding	1.69 × 10^−65^	Catalytic activity	5.48 × 10^−12^
	Organic cyclic compound binding	2.30 × 10^−46^	Ion binding	1.93 × 10^−7^
	Heterocyclic compound binding	6.98 × 10^−46^	Binding	5.32 × 10^−7^
	Protein binding	1.77 × 10^−27^	Cation binding	5.22 × 10^−5^
	Nucleic acid binding	9.34 × 10^−27^	Oxidoreductase activity	2.10 × 10^−4^
	RNA binding	1.97 × 10^−25^	Glutathione transferase activity	2.50 × 10^−4^
	Catalytic activity	6.67 × 10^−22^	Metal ion binding	3.00 × 10^−4^
	Ion binding	7.46 × 10^−22^	Transferase activity	6.70 × 10^−4^
	Carbohydrate derivative binding	4.74 × 10^−20^	Catalytic activity, acting on a protein	2.50 × 10^−3^
	Small molecule binding	1.35 × 10^−19^	Protein binding	2.80 × 10^−3^

**Table 2 ijms-25-09895-t002:** The endocrine-related gene ontology terms for upregulated and downregulated DEGs from *D. magna* chronically exposed to TNT.

GO Term Category	GO Term Description	FDR
Biological process	Organic substance biosynthetic process	5.44 × 10^−9^
	Developmental process	2.60 × 10^−4^
	Cellular process involved in reproduction in multicellular organism	4.60 × 10^−4^
	Anatomical structure development	5.10 × 10^−4^
	Reproductive process	5.60 × 10^−4^
	Female gamete generation	2.50 × 10^−3^
	Organic substance transport	7.50 × 10^−3^
	Developmental process involved in reproduction	9.40 × 10^−3^
	Multicellular organism development	1.63 × 10^−2^
	Meiotic cell cycle process	1.79 × 10^−2^
	Germ cell development	3.46 × 10^−2^
	Meiotic nuclear division	3.88 × 10^−2^
	Female meiotic nuclear division	4.15 × 10^−2^
Cellular Component	Meiotic spindle	1.05 × 10^−2^
Molecular function	Organic cyclic compound binding	1.41 × 10^−43^

## Data Availability

Data are contained within the article.

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
