# Peer review of "The Chronic Toxicity of Endocrine-Disrupting Chemical to Daphnia magna: A Transcriptome and Network Analysis of TNT Exposure"

_ijms, 2024, doi:10.3390/ijms25189895_

Round 1

Reviewer 1 Report

Comments and Suggestions for Authors

This manuscript describes gene expression profiling of Daphnia magna after exposure to TNT, an EDC. The authors found that TNT exposure disrupts the Wnt signaling pathway, causing reproductive and developmental abnormalities in D. magna.

The study was well-conducted and the results presented were generally sound. However, the authors need to clarify the purpose of this study.

The important key position of D. magna in the aquatic food chain is not relevant to revealing genetic changes in Daphnia due to TNT exposure. If the authors focused on the position of the food chain of D. magna was an issue, the changes in TNT accumulation in D. magna after TNT exposure and the metabolism of TNT in D. magna should be considered to clarify the effects of TNT accumulation on other organisms that prey on D. magna.

To focus on the genetic changes in D. magna due to TNT exposure as shown in this study, a reason should be presented that D. magna can be treated as a representative aquatic organism.

In addition, there are some points which the authors should address as follows;

1.     Is it realistically possible for aquatic organisms to be exposed to TNT? What are the concentrations of TNT in typical aquatic environments? Can the authors provide examples of rivers, lakes, and ponds that have been contaminated by TNT? And just as importantly, have any human injuries from TNT exposure been reported?

2.     In Section 4.1, what was the basis for determining the amount and duration of TNT exposure?

Reviewer 2 Report

Comments and Suggestions for Authors

General comments:

The paper is fairly straightforward and I only have a few general and specific comments. The in-line citations for references need to be fixed. Many details are missing from the methods. Figure 1 needs some attention. A combined results and discussion may serve your paper very well.

Specific comments:

Lines 12-13: can delete ‘endocrine system homeostasis, adversely affecting’ to reduce word count

Lines 14-15: It’s been a while since I’ve been steeped in the literature, but municipal wastewater is also a major source.

Line 16: ‘resulting from chronic exposure’ is redundant

Line 40: mentioning where, when, and how often TNT is used would help this paragraph.

Lines 238-270: methods are in the wrong spot

Line 241: was this a static or renewal exposure? How many organisms did you expose? Did you include a control exposure?

Line 246: this package requires a citation

Line 248: how? Does the STRING database/website include the software to complete these analyses?

Line 254: why 0.4?

Figure 1: These figures are difficult to read. I’m not entirely sure how to make them more legible, but as a start I suggest using ovals filled with blue (rather than only a blue halo) to denote downregulation and red to denote upregulation (with grey lines connecting the genes); saturation can denote the degree of change. Also, I think you can probably move 1D to its own figure to free up some space to make the other sub-figures more legible. Another option is also to highlight particular genes in some way which do or do not conform to what you expected to see (but really to make the things you highlight in the written results and the discussion very apparent).

Line 171-174: did you expect to see this?

Line 183-184: can we actually understand chronic toxicity from a 21 day exposure?

Lines 226-237: fair enough. But I would also potentially add that you looked at a very restricted exposure scenario. And while replicating all possible scenarios is not feasible, there are also additions you could make to the testing protocol which may also help us all better understand the effects of TNT: tracking gene expression both during those 21 days and afterwards to look at persistence and/or recovery. I know this adds more work, but still. Adding this type of information will help us prioritize compounds for either further study or more serious options, like regulation (although in fairness I will also admit that I don’t think specific regulations will be able to keep pace – but I digress).

Comments on the Quality of English Language

mostly fine. There are a few parts where the sentences can be made less complicated. I have highlighted two examples in my specific comments.
